# Assessing Brigada Digital de Salud Audience Reach and Engagement: A Digital Community Health Worker Model to Address COVID-19 Misinformation in Spanish on Social Media

**DOI:** 10.3390/vaccines11081346

**Published:** 2023-08-09

**Authors:** Elizabeth L. Andrade, Lorien C. Abroms, Anna I. González, Carla Favetto, Valeria Gomez, Manuel Díaz-Ramírez, César Palacios, Mark C. Edberg

**Affiliations:** 1Department of Prevention and Community Health, Milken Institute School of Public Health, George Washington University, 950 New Hampshire Ave, NW, Washington, DC 20052, USA; lorien@gwu.edu (L.C.A.); agonzalez985@gwmail.gwu.edu (A.I.G.); cmfavetto@gwmail.gwu.edu (C.F.); valeriamgomez@gwmail.gwu.edu (V.G.); medberg@gwu.edu (M.C.E.); 2La Clínica del Pueblo, 2831 15th St, NW, Washington, DC 20009, USA; mdiazramirez@lcdp.org; 3Proyecto Salud, 11002 Veirs Mill Rd, Silver Spring, MD 20902, USA; cpalacios@proyectosalud.org

**Keywords:** COVID-19, misinformation, social media, Spanish, community health workers, vaccines, health promotion

## Abstract

U.S. Spanish-speaking populations experienced gaps in timely COVID-19 information during the pandemic and disproportionate misinformation exposure. Brigada Digital de Salud was established to address these gaps with culturally tailored, Spanish-language COVID-19 information on social media. From 1 May 2021 to 30 April 2023, 495 Twitter, 275 Facebook, and 254 Instagram posts were published and amplified by 10 trained community health workers. A qualitative content analysis was performed to characterize the topics and formats of 251 posts. To assess reach and engagement, page analytics and advertising metrics for 287 posts were examined. Posts predominantly addressed vaccination (49.45%), infection risks (19.12%), and COVID-related scientific concepts (12.84%). Posts were educational (48.14%) and aimed to engage audiences (23.67%), promote resources (12.76%), and debunk misinformation (9.04%). Formats included images/text (55.40%), carousels (27.50%), and videos (17.10%). By 9 June 2023, 394 Facebook, 419 Instagram, and 228 Twitter followers included mainly women ages 24–54. Brigada Digital reached 386,910 people with 552,037 impressions and 96,868 engagements, including 11,292 likes, 15,240 comments/replies, 9718 shares/retweets, and 45,381 video play-throughs. The most engaging posts included videos with audio narration, healthcare providers, influencers, or music artists. This community-based model to engage Spanish-speaking audiences on social media with culturally aligned content to counter misinformation shows promise for addressing public health threats.

## 1. Introduction

U.S. Spanish-speaking populations experienced major gaps in timely, accurate COVID-19 information throughout the pandemic [1,2,3,4,5,6], combined with disproportionate exposure to COVID-19 misinformation, conspiracy theories and hoaxes, and targeted disinformation efforts on social media platforms [7,8,9,10]. Swells in COVID-19 misinformation and growing mistrust during the pandemic prompted healthcare providers, community leaders, and government agencies to coordinate outreach and education initiatives, yet efforts to reach Spanish-language audiences through trusted, culturally and ethnically concordant messengers often lagged behind [2,4,11].

An information vacuum in the early days of the pandemic gave way to an “infodemic,” whereby information and media environments were flooded with COVID-related content, which included evolving messaging, conflicting messages by authorities, and the politicization of government mitigation and response measures [12,13,14,15]. This created ideal conditions for the emergence of rumors and misinformation without adequate mechanisms in place to flag and remove misleading Spanish-language content from social media platforms [16,17,18]. Even with those mechanisms in place, efforts to counter misinformation have not kept pace with the quantity of false COVID-19 information circulating online in Spanish [19,20,21]. The increased vulnerability to misinformation among Latino individuals with lower literacy and health literacy levels has also been described [4,22,23]. This confluence of factors, combined with anti-immigrant and sociopolitical rhetoric [24,25], fueled growing Latino community uncertainty and mistrust of public health agencies, government institutions, scientists, and health professionals [22,26,27].

Local and national efforts to reach Latino audiences with COVID-19 information to increase vaccination have been described, and strategies have included outreach by healthcare providers and local community-based organizations through pop-up clinics and in-person outreach activities, as well as messaging through traditional media outlets [2]. With U.S. Latinos increasingly using social media as an important source of health information and as a tool for information sharing [28,29], there has been an emphasis on examining the influence of this digital environment on COVID-19 social norms, perceptions, and attitudes, especially related to vaccination [30,31,32,33,34,35]. Some efforts to reach Latino audiences with COVID-19 information have employed the use of social media [2,36,37], sometimes as part of a multi-media, multi-level campaign strategy [16,29,30].

At the same time, numerous studies and efforts have also outlined the importance of community-based strategies that leverage trusted, culturally aligned messengers to disseminate COVID-19 information, including health promoters, peer vaccine ambassadors, and community leaders [2,3,6,37,38,39]. For example, the Sin Duda campaign in Maryland by Shah and colleagues (2023) aimed to promote COVID-19 vaccines to Latino adults on social media while also connecting them to community-based vaccine and testing services [34]. To increase trust, some efforts have prioritized messaging by healthcare providers, such as the dissemination of videos featuring physicians on social media [2,30,40], while others have included messaging by influencers and celebrities to increase audience reach and engagement [28,29,30,41]. A study by Bonnevie and colleagues (2023) implemented a social media campaign to reach black women to prevent low birth weight during the COVID-19 pandemic, which incorporated videos with black physicians and partnering with local influencers [42].

However, there is a paucity of work examining the application of community-based approaches for Spanish-language COVID-19 messaging and outreach to social media networks, in particular as strategies to address misinformation.

The Brigada Digital de Salud was established in May of 2021 to address the proliferation of COVID-19 misinformation in Spanish on social media by leveraging community health workers (CHWs) as trusted sources of health information. The Brigada Digital aims to disseminate credible, science-based, and culturally relevant COVID-19 information in Spanish, including through CHW social media networks. This study characterizes the content disseminated in social media posts using the novel Brigada Digital model and presents results from a process evaluation that sought to answer two research questions: (1) What level of audience reach can be achieved using the Brigada Digital model for digital health promotion? (2) What level of audience engagement can be achieved using this model, and what post features are most engaging for a general audience of Spanish-speaking Latino adults? The study results are reported following 2 years of campaign implementation, and recommendations are provided regarding future projects that aim to replicate this digital health promotion model to engage social media audiences with COVID-19 or other health messaging.

## 2. Materials and Methods

### 2.1. Development of the Brigada Digital Model and Collaborators

The Brigada Digital was launched in May of 2021 in response to the dearth of timely, credible, and culturally relevant COVID-19 information in Spanish and the overabundance of misinformation on social media platforms. Adapted from the George Washington University (GW) Health Communication Volunteer Corps concept [43], which coordinated social media fact-checking and messaging by public health students on COVID-19 prevention and vaccination, the Brigada Digital effort leveraged community-based clinics and CHWs as trusted messengers for the outreach and engagement of Latino audiences in Spanish. The long-term objective of this initiative was to address COVID-19 disparities by increasing prevention measures, testing, and vaccine uptake among Latinos in the Washington metropolitan area through improving access to quality, audience-tailored information from trusted messengers.

The Brigada Digital effort was funded by the National Institutes of Health Community Engagement Alliance Against COVID-19 Disparities initiative (NIH-CEAL) and the GW Institute for Data Democracy and Politics. The Brigada Digital model was developed in collaboration with two community-based clinical partners in Washington, DC, and Maryland—La Clínica del Pueblo and Proyecto Salud. A partnership was also formed with Dr. Elmer Huerta [44], host of the weekday Spanish-language radio program, *Consultario Comunitario*, on Radio America 1540 AM, to support cross-platform COVID-19 messaging, and with a potential audience of over 1.15 million Latinos in the Washington, DC metropolitan area. This paper discusses the development and implementation of Brigada Digital from 1 May 2021 to 30 April 2023, which included the conceptualization of the Brigada Digital model and CHW member role, the recruitment and training of CHWs, message frame and social media content generation, the publication of posts across the Facebook, Instagram, and Twitter platforms, audience engagement strategies, and page/profile management.

### 2.2. Brigada Digital Member Role Development, Recruitment, and Training

A total of 10 CHWs, including health promoters and community health educators, were recruited in October 2021 by our two community-based partners, La Clínica del Pueblo and Proyecto Salud, to be trained as Brigada Digital members who would disseminate Spanish-language messaging with their social media networks and in local, public Facebook groups. They were identified from within the organizations’ cadres of health outreach workers based on their interest in the Brigada member role, fluency in Spanish, the use of at least one of the social media platforms (Facebook, Instagram, Twitter), and viewpoints in support of COVID-19 vaccination. Brigada members received 2 h of training in November 2021 on the Brigada Digital model, the process for sharing posts across social media platforms, considerations for digital health promotion and audience engagement, and how to navigate conversations involving COVID-19 misinformation. Trained members were asked to follow Brigada Digital pages, share weekly posts with their networks and in public groups, engage with network members to answer questions and provide resources, conduct in-person outreach to expand their digital networks, and submit weekly reports on their digital health promotion activities. Brigada Digital CHWs spent approximately 4–5 h per week dedicated to these activities.

### 2.3. Theoretical Basis and Messaging Strategies

Initially, a message frame was developed, informed by a cultural adaptation of the theory of planned behavior (TPB), whereby COVID vaccination depended on the intention to vaccinate, which was influenced by beliefs about vaccination, social norms, perceived control to vaccinate, and attitudes about vaccination [45]. TPB was operationalized to include underlying cultural values and culturally normative social norms, including expectations of social closeness (i.e., *personalismo*) and the importance of family relationships (i.e., *familismo*), which have been widely applied in health promotion and communication interventions with Latinos [46,47,48,49,50,51].

Given the rapidly evolving nature of the pandemic and the need for specific, sometimes unpredictable, messaging to be developed quickly, weekly content development for Brigada Digital also evolved to incorporate messages that were responsive to contextual factors and salient phenomena in the information environment. For example, when public health policies and guidelines changed, the results of a new vaccine trial were released, child vaccines were approved for certain age groups, free COVID-19 tests were newly available, or there was news coverage of a new variant, messaging would immediately pivot to prioritize the dissemination of this information (See Figure 1).

We also synchronized messaging to align with topics of social relevance, such as upcoming holidays, the start of the school year, and current events (i.e., World Cup, holiday travel, inflation). Furthermore, to promote audience engagement, posts incorporated humor where appropriate, easily recognizable references to Latino culture and experiences, popular and trending Latin music artists and songs, and connections to cultural celebrations (See Figure 2a,b).

Messaging was also responsive to online conversations that included COVID-19 misinformation, whereby posts were developed to directly address incorrect information that was widely circulating on social media platforms. Our team used a number of tactics to counter COVID-19 misinformation on social media, including disseminating credible, evidence-based information with links to original information sources; directly addressing/debunking specific pieces of misinformation; dissemination using trusted sources as messengers (CHWs) and with trusted sources portrayed in content (i.e., Latino pediatricians or health professionals); and tailoring content to be accessible by Spanish-language audiences and those with varying levels of literacy and science literacy [52,53,54].

### 2.4. Culturally and Contextually Tailored Content, Formats, and Accessibility

Approximately 2–3 COVID-19-related posts were published per week on average across accounts on all platforms. Spanish-language content was developed to communicate across the following topic domains: COVID-19 transmission and risk/severity in adults and children; the science of COVID-19 variants and immunity; COVID prevention measures (masking, types of masks, social distancing, ventilation, handwashing) and testing (when to test, obtaining free tests); COVID-19 vaccine and booster contents, safety and efficacy for adults, children, and pregnant/breastfeeding individuals; overcoming barriers to obtain vaccines and boosters; and COVID-19 treatment options.

Content was developed to be delivered in a range of formats, from narrated slide carousels and animated images with text to video interviews and observational tutorials. The intent was to create engaging digital content that resonated with Latino audiences and was accessible to individuals with diverse educational backgrounds and health literacy levels [55,56]. Video content portrayed Latino physicians, CHWs, public health professionals, and community leaders, with a few including influencers, music artists, or celebrities. Complex scientific concepts were simplified and explained, visual illustrations were used, text length was minimized, font size was increased, and some longer texts were audio-narrated in Spanish (See Figure 3a,b).

Brigada Digital content also acknowledged the multiple structural, socioeconomic, and political factors that shape options for diverse Latino communities with respect to implementing COVID-19 prevention and mitigation recommendations. Standard messages, even when translated into Spanish, do not always take such factors into account, thus diminishing their potential impact. Messages were developed to be realistic given potential contextual barriers and focused on feasible behavior changes within these contexts.

### 2.5. Implementation Process—Content Development, Dissemination, and Audience Engagement

Given the challenges of communicating about COVID-19 in the pandemic context, including rapidly evolving (and often conflicting) messages and complex scientific topics, we wanted to ensure that any messaging coming from the Brigada Digital had been thoroughly researched, fact-checked, and vetted, was accurate and up-to-date with the last public health guidance, was consistent with the current scientific evidence, and most importantly, was (accurately) simplified and accessible to broader community audiences. We also wanted to ensure that messaging disseminated by Brigada Digital CHW members to their social networks was concordant with messages conveyed by the principal Brigada Digital accounts on Facebook, Instagram, and Twitter. In order to support digital community outreach with consistent, up-to-date COVID-19 information in this saturated media environment, we centralized the process of weekly message and content curation, and all content was created by a small bilingual, bicultural research team of three individuals, including a university faculty member with health promotion and public health crisis and emergency risk communication expertise and two public health students from the GW Milken Institute School of Public Health. Messaging and content development were guided by the aforementioned theoretical framework and message frame, and with input from community partners and Brigada Digital CHW members.

Each week, the team met to discuss the following to guide content development: (1) the latest COVID-related news developments, scientific updates, and changes to public health guidance; (2) predominant misinformation topics that had emerged or were being actively discussed on social media; (3) COVID-specific concepts that were relevant at that time and were often misunderstood (i.e., how to correctly wear masks, when to test, eligibility for booster doses); (4) upcoming social or cultural events/days (i.e., holidays, International Women’s Day, Hispanic Heritage Month); and (5) trending topics in current events or popular culture with which we could align our content or messaging strategy (i.e., Minions movie, music artist Daddy Yankee announcing retirement, inflation). After the identification of key topics to address each week, messages were developed, target audience segments were identified (i.e., young adults, parents of children under 18), and creative post concepts were decided on, including the media/format to be used (i.e., static or animated post, video, tutorial, narrated carousel). We also decided on communication tactics (i.e., use of anecdotes, emotional appeals, culturally informed alignment with family values or community solidarity) and audience engagement strategies (i.e., storytelling, use of music, humor, celebrity).

The majority of social media content, including the main post, caption, and source reference links, was developed by our team in Canva [57]. Tools available in Canva were relatively low-cost and user-friendly and facilitated the elaboration of post content, including branded visual elements and media. Given that Canva did not have features that directly supported Spanish language content generation, posts were developed in English and externally translated using Spanish that avoided regional colloquialisms or included terminology that would be widely understood by audiences of different backgrounds. After review and completion, posts were downloaded as high-resolution image or video files and then uploaded to Brigada Digital platforms with corresponding captions and source links. To coordinate the content curation across the team, we used the organizational tool, Trello, which enabled us to monitor post development from concept to completion and integrate appropriate team members at each stage [58]. To begin this process, we added post concepts from our weekly meeting, which were then approved by the principal investigator, and claimed by a lead developer. Once a draft post was developed, the concept was tagged as ready for review, and once reviewed, it was tagged as ready for posting, and then boosting. This process incorporated multiple review checkpoints and ensured high-quality content prior to publishing.

The Brigada team developed content on a weekly schedule, whereby our team would aim to have content published on Mondays, Wednesdays, and Fridays of every week. We deviated from this cycle in the case of breaking news, such as the announcement of a shift in COVID-related public health guidance, in which case we would aim to share this information as soon as possible. We sought to have at least one post per week that was educational, one that included news updates/explained scientific news, and one that was intended primarily for engagement. All content was cross-posted on our Facebook page and Instagram and Twitter profiles. The majority of posts were boosted with approximately USD 10–15 in paid advertising per post for a 5-day period on the Facebook and Instagram platforms. The selection of audience demographic segments varied by post depending on the post’s content and intended audience (i.e., young adults, parents of children under 18) but targeted segments were always among adults ages 18–65 living in the U.S. Most post advertisements also targeted the audience segment of expats from countries in Central and South America as proxy for reaching Latino, Spanish-speaking individuals.

Posts on our principal accounts were also shared by Brigada Digital CHW members to their social media networks throughout the week and to public Spanish-language Facebook groups in the Washington, DC area. Across the 10 CHWs, they used individual accounts, including 1 Twitter, 10 Facebook, and 7 Instagram accounts, to disseminate Brigada Digital content. CHWs connected with their networks to elicit engagement, answer questions, and refer individuals to additional resources, services, and events. Each Brigada member submitted a weekly report via Google Forms, in which they reported on their number of posts, audience engagement, and relayed any questions or challenges. We also created a WhatsApp group chat to cultivate a space where Brigada CHWs could connect with our team and each other to collaborate, exchange ideas, and seek assistance. This also served as a channel through which CHWs could suggest ideas for post content based on interactions with their clients and audience. This enabled us to incorporate diverse, community-based perspectives to ensure that content resonated with our audiences. Posts were also shared by La Clínica del Pueblo on the organization’s Facebook account, which has 5200 followers.

In addition to weekly posts, we also created additional audience engagement opportunities. For example, in early 2022, we collaborated with Dr. Elmer Huerta, the host of the Spanish-language radio program, *Consultario Comunitario*, to arrange a series of six radio interviews with pediatricians and Brigada Digital member health promoters from our community-based clinical partner organizations. The interviews were intended to convey key messaging about COVID-19 vaccination for adults and children and to correct predominant misinformation about the vaccines. The interviews were aired live on the radio program and simultaneously livestreamed on the Brigada Digital Facebook page, with brief interview clips with concise messaging later disseminated as posts across all Brigada platform accounts.

Our team continually monitored all Brigada Digital accounts to reply to direct messages, acknowledge comments, respond to inquiries, provide links to additional resources, and flag comments that were inappropriate or contained misinformation. Generally speaking, in cases where misinformation was identified in comments, our overall approach was to acknowledge the user’s perspective or concern, find common ground when possible (i.e., we also value transparency, or as parents, we also prioritize the health of our children), maintain a respectful and helpful tone, and request that the user provide links to support their claims so we could assess the reliability of their information sources. Our team also responded to comments/replies containing misinformation within 24 h or less and never left any comment/reply containing misinformation without a response from our team. To counter misinformation, we researched and fact-checked all claims and formulated a response with links to credible sources of information or prior relevant Brigada Digital posts (always cited) to support the counter-claims. We initially aimed to maintain all comments containing misinformation as publicly viewable, but with our counter-responses so as to maintain the appearance of transparency and open dialogue, as well as to provide a “teachable opportunity” to others regarding how to identify misinformation and which information sources should be trusted instead. There were, however, a limited number of cases where we had to remove/hide comments or block certain users due to the aggressive nature of their comments or the large quantity of their comments that our relatively small research team could not effectively manage without compromising the integrity of our page.

### 2.6. Data Collection and Analysis

A total of 275 Facebook posts, 254 Instagram posts, and 495 Twitter tweets were published between 1 May 2021 and 30 April 2023 on the three main Brigada Digital accounts on these platforms. Within these totals, 251 posts were original educational/engagement posts created by Brigada Digital, which were supplemented by shared/retweeted COVID-19-related content from other accounts, and a small number of posts were for the recruitment of evaluation study participants (discussed elsewhere).

In order to characterize Brigada Digital’s post content, we completed a qualitative content analysis for the original 251 posts published during this same time frame using categories for post topic, media/format type, and post purpose. A priori categories were selected by one of the PIs who oversaw content development and were informed by the Brigada Digital message frame that guided the selection of educational topics (i.e., COVID-19 vaccination, testing, masking), and the weekly content development process in which we decided post formats (i.e., slide carousels, videos, images), purposes (i.e., address specific misinformation, engage audiences, or providing a COVID-related news update) and engagement features to be used for each post (i.e., animation, music, audio narration). All posts in the Brigada Digital Facebook feed for the 2-year period (which were cross-posted across platforms) were reviewed by one coder, and a categorical code was applied for post topic, primary and secondary purpose, and format/media types. Codes for each category were entered into an Excel spreadsheet, and then frequencies for each category were produced using the STATA software, version 17 (College Station, TX).

Furthermore, to characterize audience members, we also used insights available in the Meta Business Suite for the main Brigada Digital Facebook and Instagram accounts. These insights provided basic user demographic profiles for Brigada followers on these platforms. Frequencies for audience age groups, gender, and country are reported from platform analytics as of 9 June 2023.

Research Question 1: To assess audience reach, we used Facebook, Instagram, and Twitter page/profile insights for the three main Brigada Digital accounts (representing all posts, including both original content posts and posts shared from other accounts) as well as analytics from the Ad Reports tool available through the Meta Business Suite (representing only boosted posts placed on Facebook and Instagram). Cumulative summary metrics are reported for the period of 1 May 2021 through 30 April 2023. The following metrics were used to assess reach: page/profile followers, reach, page/profile visits, and impressions (See Table 1 for metrics calculations by platform).

Research Question 2: To assess audience engagement, similar to audience reach, we used Facebook, Instagram, and Twitter page/profile insights and analytics from the Meta Business Suite Ad Reports tool for the three main Brigada Digital accounts. Cumulative summary metrics are reported for the period of 1 May 2021 through 30 April 2023, and the following metrics were used: post/tweet engagements, likes, comments/replies, shares/retweets, video play-throughs, detail expands, and link clicks (See Table 1).

In order to assess the most engaging Brigada Digital content, we identified posts with the highest engagement and examined the content to identify patterns in shared post characteristics that could potentially explain this higher audience engagement. We created a report for Facebook and Instagram posts using the Meta Ad Reports tool and identified posts with the highest values for the Post Engagement metric, which is a composite measure of post shares, reactions, saves, comments, likes, interactions, 3 s video plays, photo views, and link clicks. The six posts (with an educational purpose) with the highest Post Engagement values were identified, as were the posts with an engagement purpose and the highest Post Engagement values. The topics, formats, and features of these posts, as well as communication tactics used (i.e., tagging social media influencers, selected advertising audience, timing of post), were then examined to identify patterns in post characteristics that may potentially explain higher levels of audience engagement.

## 3. Results

### 3.1. Content Overview

Original Brigada Digital posts (n = 251) covered a wide variety of topics related to COVID-19, and there was also a small subset of topics that were not directly related to COVID-19, which were typically included to update audiences with other related news (i.e., Mpox or bird flu outbreaks) or to engage audiences in current events (i.e., sympathy for Hurricane Fiona victims or solidarity with Ukraine) (See Table 2).

Of the 251 original posts assessed for content, 115 posts addressed more than one topic, resulting in the topics listed above being covered a total of 366 times. For example, a post may have primarily discussed risk of infection with a new COVID-19 variant (primary topic) and promote vaccination as the best way to protect oneself (secondary topic). The topics most commonly included in Brigada Digital posts were: COVID-19 vaccination, COVID-19 risks, and the science of COVID-19. Approximately half of post topics (49.4%) addressed COVID-19 vaccination and booster doses for adults and children, including vaccine safety and efficacy. About one-fifth of topics (19.9%) conveyed COVID-19-related risks, such as the risk of transmission and the risks associated with infection for different population subgroups, and 12.8% of topics explained the scientific concepts behind COVID-19 transmission and virus variants.

Brigada Digital posts were developed with different purposes in mind, ranging from educating or promoting protective behaviors, addressing specific misinformation, updating audiences about changing public health guidelines or revised vaccine eligibility criteria, connecting people to resources and events, or simply engaging audiences (See Table 3).

A total of 125 posts were developed with more than one purpose. For example, a post may have had the primary purpose of educating audiences, while also debunking misinformation about the COVID-19 vaccine that was circulating online. The most common post purpose was health education and promotion regarding COVID-19 risk and mitigation, with almost half of the posts having this purpose (48.1%). The next most common post purposes were to engage/entertain audiences with content related to COVID-19 (14.1%) or current events (9.6%), link individuals to resources, services, or vaccination/educational events (12.8%), and to address COVID-19 misinformation (9.0%).

Brigada Digital posts were also developed to be varied in terms of formats and media types (See Table 4).

Approximately half of the posts consisted of images with text (55.4%), including 4.8% with animation and/or music. Just under one-third of posts were carousels (27.5%), with 2.8% containing music or Spanish audio narration. Finally, almost one-fifth of posts (17.1%) were videos, with 2% of videos including music.

### 3.2. Brigada Digital Content Advertisements

A total of 295 posts were boosted during the 2-year period, and half of the advertisements had the specific objective of increasing post engagement (n = 149, 50.5% of ads), video views (n = 106, 35.9% of ads), or link clicks (n = 40, 13.6% of ads). Approximately half of the advertisements were placed on Facebook (n = 147, 49.8% of ads) and almost half on Instagram (n = 140, 47.5% of ads), with the remainder placed on Messenger or elsewhere (n = 8, 1.7% of ads). The total spent on post advertisements during the 2-year period was USD 2813, with an average of USD 9.54 spent per post.

### 3.3. Research Question 1: Brigada Digital Audience Reach

Regarding the level of audience reach achieved using the Brigada Digital model, as of 9 June 2023, Brigada Digital had 394 Facebook, 419 Instagram, and 228 Twitter followers. Across the three platforms, the Brigada Digital de Salud effort was able to reach 386,910 people, receive 9554 page/profile visits, and achieve a total of 552,037 impressions during the 2-year period (See Table 5).

Approximately three-quarters of Facebook and Instagram audience members were women, with women ages 25–54 representing 76.5% of Facebook and 84.7% of Instagram followers (See Table 6).

Audience members were also predominantly from the U.S. and Puerto Rico (90.8% and 73.5% for Facebook and Instagram, respectively). Information on Twitter followers of the Brigada Digital account was unavailable.

### 3.4. Research Question 2: Brigada Digital Audience Engagement

Regarding the level of audience engagement achieved with the Brigada Digital model, during the 2-year period, the Brigada Digital de Salud effort was able to elicit 96,868 engagements, which included 11,292 likes, 15,240 comments/replies, 9718 shares/retweets, 45,381 video play-throughs, 590 detail expands, and 741 link clicks across the three platforms (See Table 7).

Additionally, we sought to examine Brigada Digital’s most engaging posts to identify patterns in common post features that may have contributed to higher engagement. A summary of the most engaging Facebook and Instagram posts is presented in Table 8. These posts include the top six most engaging posts with a primary educational purpose and the top six most engaging posts with a primary purpose of eliciting audience engagement.

The 12 most engaging posts covered a variety of topics, including COVID-19 vaccine safety, efficacy, and cost, booster doses, COVID-19 tests, and masks; however, one-quarter of the most engaging posts addressed child vaccination, specifically. Four out of the six most engaging educational posts addressed common COVID-19 vaccine misinformation, specifically as it relates to child vaccination (i.e., that a multi-dose vaccine series was unusual, that the vaccine was not tested adequately). Three out of the six most engaging posts that were designed specifically to increase engagement promoted free resources for COVID-19 mitigation (i.e., how to obtain free tests and masks). All 12 of the most engaging posts were delivered in video-based formats, including video interviews, tutorials, narrated slides, animated images, or videos with text and music. Eleven of the twelve posts included audio, with five posts including music (three of which were by celebrity artists), three posts including an audio-narration feature in Spanish, two being delivered in an interview format, and one containing comedic narration by a social media influencer. A top-performing post that did not include audio featured an animated gif of a well-known Latino actor and comedian. Six of the twelve posts portrayed widely-recognized social media content creators, popular music artists, and actors, and two posts included renowned and trusted health experts.

## 4. Discussion

### 4.1. Brigada Digital Model

The Brigada Digital model for digital community outreach and health promotion was well received by audience members, partners, and CHWs, and shows considerable promise as an approach that leverages strong, extensive Latino community networks while having the adaptability and reach of digital platforms. Most importantly, this approach reaches individuals on platforms where Spanish-speaking audiences are interacting with large quantities of misinformation, and efforts to counter this misinformation by trusted, culturally aligned sources are lacking. The Brigada Digital model for digital community-based health promotion also has the potential to address other important health disparities and public health priorities beyond COVID-19.

The approach we used for content generation was guided by a theoretical framework and overall message frame, yet was flexible enough to determine messaging priorities on a weekly basis. This allowed for a highly adaptable strategy that could shift messaging rapidly to align with evolving public health guidelines, deliver timely information with breaking news and scientific developments, and capitalize on the attention surrounding novel or trending topics. At the same time, this approach also required a great deal of continual investment of time and resources from a relatively small research and content development team. Given the nature of the pandemic digital information environment and the polarization of COVID-19-related topics, there were added demands on the team to maintain audience trust by delivering the highest quality content that was thoroughly researched, accurately translated, and adequately adapted for accessibility. Future efforts using this model should consider these factors when planning the project’s scope and determining personnel inputs.

One continued barrier was the limited availability of additional informational resources in Spanish to which we could refer audience members. This also presented challenges in substantiating scientific claims about COVID-19 when relevant literature or reports were unavailable in Spanish. If public health and scientific communities aim to build trust with language minority communities, there need to be more credible, science-based information sources that are appropriate for individuals with diverse literacy, science literacy, and health literacy competencies. Spanish language information available on government websites often consists of dense and lengthy text with few images and reading levels that were too high for general audiences.

### 4.2. Research Question 1: Brigada Digital Audience Reach

We anticipated that Facebook and Instagram platforms would be the most conducive for reaching the intended audience, and these platforms did prove to be appropriate channels, similar to other studies of social media campaigns reaching Latino adults in the U.S., including those related to dissemination of COVID-19 information [16,30,37,58]. Initially, Brigada Digital content was predominantly disseminated on Facebook, Instagram, and Twitter, and while the results are not presented here, we did expand outreach to include the TikTok and LinkedIn platforms. As also suggested by other studies, platforms such as YouTube and WhatsApp should also be considered for future efforts, which have proven to be very popular among Latino audiences [16,30,37].

The Brigada Digital effort was able to achieve substantial reach across these platforms, which was further supported by CHW outreach, including to local and public Facebook groups and by paid social media advertising. The level of audience reach observed in this study is comparable to the reach achieved by other initiatives of a similar scale. For example, the previously referenced Sin Duda campaign by Shah and colleagues (2023) implemented two intensive social media campaign modules lasting 6–8 weeks each in duration and featuring 4–6 new advertisements from March 2021 to March 2022, reaching a total of 305,122 people through Facebook and Instagram advertisements, and resulting in 9607 website visits. However, the advertising budget for this effort remains unclear, and although this campaign coupled social media outreach with a connection to community-based organization services, it did not have a CHW digital outreach component, and the metrics for audience engagement are not provided [34]. Additionally, a campaign study by Bonnevie and colleagues (2023) was of a similar scale to the current study, and shared many similarities as well, including the theoretical basis, platforms used, and branded social media content with similar formats, such videos with black physicians. Instead of 10 CHWs, this campaign partnered with 71 local influencers to expand organic reach, and with an advertising budget of USD 200 per month (six times the current study’s advertising budget), the campaign was able to reach 19,875 individuals, gain 1234 followers, and achieve 805,437 impressions [42].

While the current study achieved audience reach that was comparable to these efforts, two important considerations should inform future efforts so as to maximize audience reach: (1) selection of CHWs and (2) strategies to increase organic reach. Given that research has shown individuals to be more likely to trust information sources they already know and trust at the outset of a crisis—in this case, the pandemic [59,60,61,62,63]—we recommend that future studies seek to identify CHWs with existing and extensive social media networks that include target audience members. Furthermore, as described by other studies, the segmentation of audiences targeted by social media advertisements [64] and partnerships with social media individual and organizational influencers and celebrities can further increase organic reach [28,30,58].

### 4.3. Research Question 2: Brigada Digital Audience Engagement

The Brigada Digital effort resulted in considerable audience engagement, which was supported by incorporating cultural elements, leveraging current events and breaking news topics, and using humor to make COVID-19 education more appealing, especially after pandemic fatigue set in. Overall, content delivered in video formats, whether as animated text/gifs or elaborate dramatizations, tended to result in more audience engagement overall than simple text with images, which is consistent with findings from other studies [16,27,59]. A novel finding was the increased engagement with posts that included an audio narration feature, in which text scrolls across the page as it is read aloud. We had primarily included this feature to increase the accessibility of content for lower literacy populations but also discovered that it seemed to boost engagement as well. If future initiatives have limited resources to produce more elaborate video content, using audio narration or scrolling/typewriter animation text features can be a strategy to consider.

Another feature of Brigada Digital posts that also tended to result in higher levels of audience engagement was information delivery by trusted messengers, such as a pediatrician or a well-known radio/tv personality and physician. This finding is consistent with other interventions and campaigns that have used a similar strategy of including video-based content featuring culturally aligned healthcare providers [2,30,59]. Additionally, the inclusion of imagery and references to famous music artists or actors, and tagging/collaborations with social media influencers, resulted in greater audience engagement in the current study. As audience engagement is closely aligned with organic reach, we concur with other studies in recommending that future interventions prioritize these strategies to increase reach and also optimize audience engagement [16,36,59].

Finally, while offering excellent reach and opportunities for audience engagement, digital health promotion outreach about COVID-19 on social media also created opportunities for “negative” engagement by individuals who were actively spreading COVID-19 misinformation and conspiracy theories online. This increased our team’s responsibility to continuously and carefully monitor post comments to identify and address misinformation in real time. Projects seeking to implement community-based digital outreach with CHWs must have sufficient personnel and capacity to monitor, fact-check, and correct misleading and false comments, which can be created by users who are very active communicators on social media and who we found to employ complex and manipulative communication tactics that can be difficult to navigate (experiences and lessons learned in this area discussed in depth elsewhere). Digital CHWs who may encounter and/or manage this kind of engagement on social media platforms should also receive enhanced training on how to recognize and respond to these tactics and effectively counter misinformation.

The findings from this study represent a process evaluation of the novel Brigada Digital de Salud model for digital health promotion with CHWs on social media, addressing an important gap in public health practice and research literature that explores how messaging by trusted CHW can reach and engage audiences in the digital space. These findings are also significant due to the COVID-19 disparities experienced by Latinos and the continued disproportionate exposure to health misinformation on social media by Spanish-language audiences. Given the harms of misinformation and the growing mistrust in science and health sciences research, it is crucial that we build community capacity to leverage these trusted networks and messengers to combat misinformation and its consequences. This model offers a promising approach that can be replicated to address both current and emergent public health challenges using a community-based strategy for digital audience reach and engagement.

### 4.4. Limitations

There are some limitations that should be considered when interpreting study results. In terms of assessing audience reach and engagement, analysis was limited to what was available from platform page/profile analytics and advertising report metrics, with advertising reports usually offering more detailed results. While Facebook and Instagram tended to provide more detailed page/profile analytics, Twitter had more severe limitations; for example, audience reach metrics were unavailable for Twitter. Furthermore, relatively limited platform-based information about audience demographics made it difficult to fully characterize Brigada Digital audience segments, and Twitter audience metrics were also unavailable. This presents challenges with regard to interpreting the generalizability of study findings to Latino and Spanish-speaking audiences; however, since women between the ages of 25 and 54 were overrepresented among Brigada Digital followers, it could be inferred that study results may not be generalizable to other gender and age subgroups. Additionally, audience reach tends to be determined to a great extent by paid advertising on platforms. Given that this effort was implemented with a relatively small advertising budget and advertisements were not placed on Twitter during the study period, results in terms of audience engagement should be interpreted with these factors in mind. Finally, this study entailed digital outreach in Spanish implemented across three social media platforms, and the results may not be comparable with outreach on other social media platforms.

## Figures and Tables

**Figure 1 vaccines-11-01346-f001:**
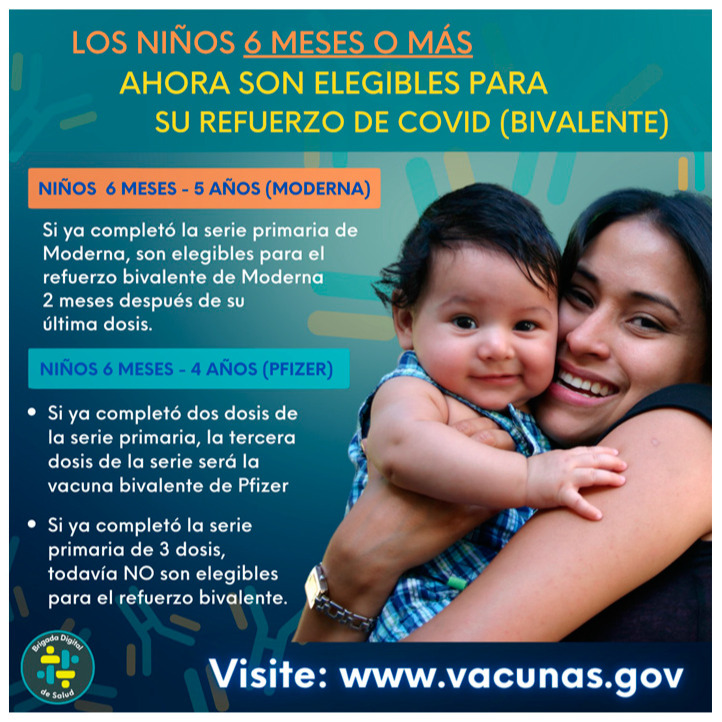
Post announcing COVID-19 vaccine availability for children under age 5.

**Figure 2 vaccines-11-01346-f002:**
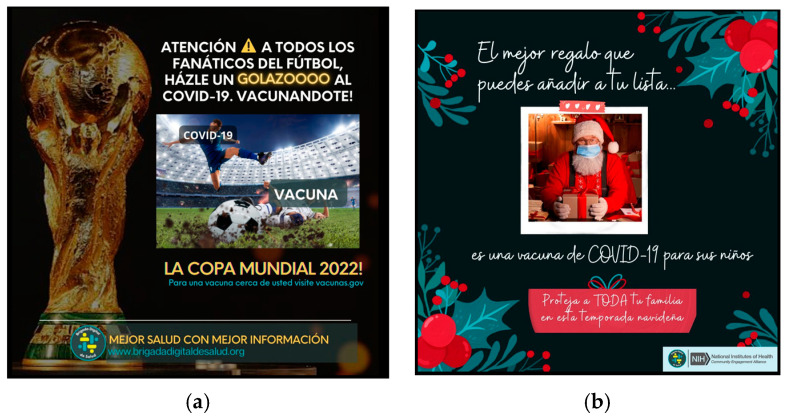
(**a**) Post engaging audience for the World Cup; (**b**) post engaging audience for Christmas holiday.

**Figure 3 vaccines-11-01346-f003:**
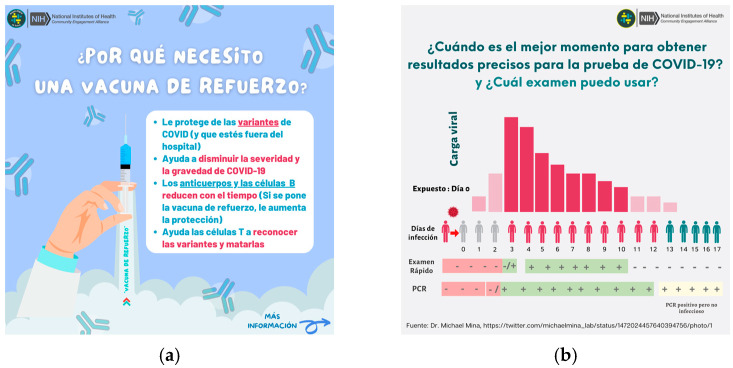
(**a**) Post educating about how booster doses work; (**b**) post educating about when to test for COVID-19.

**Table 1 vaccines-11-01346-t001:** Metrics calculations by Twitter, Facebook, and Instagram platforms.

Metrics	Twitter	Facebook	Instagram
Audience Reach			
Reach	--	Number of accounts that saw a post, story, or ad at least once (organic and paid)	Number of accounts that saw a post, story, or ad at least once (organic/paid)
Impressions	Number of times a tweet was seen	Number of times a post was seen (organic and paid)	Number of times a post was seen (organic/paid)
Page/Profile Visits	Number of profile visits	Number of page visits	Number of profile visits
Audience Engagement			
Post Engagement	Sum of likes, replies, retweets, 2 s media views, detail expands, profile clicks, hashtag clicks, link clicks, and new followers gained	Sum of post shares, reactions, saves, comments, likes, interactions, 3 s video plays, photo views, and link clicks while ad is running	Sum of post shares, reactions, saves, comments, likes, interactions, 3 s video plays, photo views, and link clicks while ad is running
Likes/Reactions	Number of tweet likes	Number of post likes	Number of post likes
Comments/Replies	Number of tweet replies	Number of post comments	Number of post comments
Shares/Retweets	Number of retweets	Number of shares to timelines, groups, or pages	Number of shares
Video Thru-plays	--	Number of times video was played to completion, or for at least 15 s	Number of times video was played to completion, or for at least 15 s
Detail Expands	Number of clicks to view more details	--	--
Link Clicks	Number of link clicks in tweet	Number of link clicks in post	Number of link clicks in post

**Table 2 vaccines-11-01346-t002:** Brigada Digital social media post topics.

Post Topic	Posts with Primary Topic (N)	Posts with Secondary Topic (N)	Total N (%)
Brigada Promotion	8	0	8 (2.2%)
COVID Science & Variants	4	43	47 (12.8%)
Masking	9	9	18 (4.9%)
Multi-prevention	1	8	9 (2.5%)
Non-COVID	19	0	19 (5.2%)
COVID—Pregnancy/Breastfeeding	1	4	5 (1.4%)
COVID Risk Communication	69	1	70 (19.1%)
Social Distancing/Air Quality	1	2	3 (0.8%)
COVID Testing	5	1	6 (1.6%)
COVID Boosters	9	13	22 (6.0%)
COVID Vaccination—Child/Adolescent	30	10	40 (10.9%)
COVID Vaccination—General	81	11	92 (25.1%)
COVID Vaccination—Safety/Efficacy	14	13	27 (7.4%)
	Total = 251	Total = 115	Total = 366

**Table 3 vaccines-11-01346-t003:** Brigada Digital social media post purpose.

Post Purpose	Posts with Primary Purpose (N)	Posts with Secondary Purpose (N)	Total N (%)
Address Misinformation	5	29	34 (9.0%)
Educational/Health Promotion	175	6	181 (48.1%)
Engagement/Entertainment	33	20	53 (14.1%)
Popular Culture/Current Events	14	22	36 (9.6%)
Link to Resource/Service/Event	11	37	48 (12.8%)
News Update	13	11	24 (6.4%)
	Total = 251	Total = 125	Total = 376

**Table 4 vaccines-11-01346-t004:** Brigada Digital social media post formats and media types.

Post Format	Total N (%)
Carousel	62 (24.7%)
Carousel—Music	3 (1.2%)
Carousel—Audio-Narration	4 (1.6%)
Image and Text	127 (50.6%)
Image and Text—Animation	6 (2.4%)
Image and Text—Music	6 (2.4%)
Video	38 (15.1%)
Video—Music	5 (2.0%)
	Total = 251

**Table 5 vaccines-11-01346-t005:** Brigada Digital audience reach metrics (1 May 2021–30 April 2023).

Metrics	Twitter (N = 493 Tweets)	Facebook (N = 275 Posts)	Instagram (N = 254 Posts)	Total (N = 1022 Posts)
Reach	--	336,427	50,483	386,910
Page/Profile Visits	134	5593	3827	9554
Impressions	37,809	431,018 *	83,210 *	552,037

* Metric for posts boosted with paid advertisements only.

**Table 6 vaccines-11-01346-t006:** Brigada Digital Facebook and Instagram followers, by gender and age *.

Platform	Facebook (N = 394)		Instagram (N = 419)	
Age Group	Women N (%)	Men N (%)	Women N (%)	Men N (%)
18–24	14 (3.6)	7 (2.0)	28 (6.7)	8 (1.9)
25–34	62 (15.7)	15 (3.8)	113 (27)	34 (8.1)
35–44	107 (27.0)	25 (6.4)	115 (27.4)	35 (8.4)
45–54	74 (18.8)	19 (4.8)	52 (12.4)	6 (1.4)
55–64	36 (9.3)	17 (4.3)	17 (4.0)	4 (1.0)
65+	12 (3.0)	6 (1.6)	6 (1.5)	1 (0.2)
Total (%)	305 (77.4)	89 (22.5)	331 (79)	88 (21)

* Audience profile as of 9 June 2023.

**Table 7 vaccines-11-01346-t007:** Brigada Digital audience engagement metrics (1 May 2021–30 April 2023).

Metrics	Twitter (N = 493 Tweets)	Facebook (N = 275 Posts)	Instagram (N = 254 Posts)	Total (N = 1022 Posts)
Post Engagements	3287	77,965	15,616	96,868
Likes	825	7969	2498	11,292
Comments/Replies	242	14,879 *	119	15,240
Shares/Retweets	411	7833 *	1474	9718
Video thru plays	--	40,251 *	5130 *	45,381
Detail Expands	590	--	--	590
Link Clicks	51	460	230 *	741

* Metric for posts boosted with paid advertisements only.

**Table 8 vaccines-11-01346-t008:** Brigada Digital most engaging Facebook and Instagram posts (1 May 2021–30 April 2023).

Post Topic	Format	Reach	Impressions	Engagement	Thru Plays	Features
Many vaccines require multiple doses, like routine child vaccines	Narrated video	6243	7165	2205	949	Addressed misinformation, audio-narrated
Recap: 2 years since first COVID vaccine administered	Video with text & music	4684	2815	1595	1209	Music, typewriter text, visual appeal
Radio host interview with pediatrician—1	Video expert interview	2892	6721	1565	273	Behind the scenes of radio show with Latino celebrity Dr. host; credible expert
Radio host interview with pediatrician—2	Video expert interview	1666	3608	1108	220	Behind the scenes of radio show with Latino celebrity Dr. host; credible expert
Six things to know about COVID vaccine safety for children	Narrated Video	2124	2319	1104	968	Addressed misinformation, audio-narrated; timed with start of school year
When to receive booster dose	Narrated video tutorial CDC tool; link to website	1737	1902	925	670	Step-by-step, narrated instructional video on how to use online CDC resource in Spanish
Did you remember Mother’s Day?	Video dramatization	22,173	25,738	6943	3125	Social media influencer portrayed; repurposed humorous video with COVID messaging; trending topic—holiday
How to obtain free COVID-19 tests	Video—animated gif; link to website	7861	10,461	4898	1553	Portrayed well-known Latino comedic actor with engaging expression; promoted free resource
Booster dose provides best protection	Image with music; link to website Engagement	7953	4633	2805	1496	Popular Latina music artist; trending topic—gossip of romantic breakup; upbeat dance music
How to order free COVID tests and masks	Video with music; link to websiteEngagement	3099	3445	1497	1274	Popular Latino music artist; music
Real-life story of boosters reducing COVID transmission	Animated text with musicEngagement	4173	4470	1294	972	Storytelling; typewriter text; music; rhyming poem
Inflation prices are high, but COVID vaccines are still free	Video with text & musicEngagement	1597	1856	826	736	Current events—price increases for popular items juxtaposed against free vaccine; upbeat music from popular artist

## Data Availability

The data presented in this study are available on request from the corresponding author. Data are not publicly available due to the current regulation on privacy.

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
