# Peer review of "Assessing Brigada Digital de Salud Audience Reach and Engagement: A Digital Community Health Worker Model to Address COVID-19 Misinformation in Spanish on Social Media"

_vaccines, 2023, doi:10.3390/vaccines11081346_

Round 1
Reviewer 1 Report
The article brings the strategy used to inform timely the U.S. Spanish-speaking populations, on COVID-19.. And Brigada Digital de Salud was established to address these gaps with culturally-tailored, Spanish-language COVID-19 information on social media. All the health related information on the Covid-19 were covered. In order to measure the efficacy of the Brigada Digital de Salud, additional indicator should be used. Table 2 shows that 49,45 % of topics were related to Covid-19 vaccination: COVID Boosters 9 13 22 (6.01%) COVID Vaccination – Child/Adolescent 30 10 40 (10.92%) COVID Vaccination – General 81 11 92 (25.14%) COVID Vaccination – Safety/Efficacy 14 13 27 (7.38%). Therefore, the up-take of Covid-19 vaccination by the US spanish population could be a good indicator the Brigada efficacy.
Author Response
Reviewer 1 Comments and Suggestions for Authors:
- The article bring the strategy used to inform timely the U.S. Spanish-speaking populations, on COVID-19. And Brigada Digital de Salud was established to address these gaps with culturally-tailored, Spanish-language COVID-19 information on social media. All the health-related information on the Covid-19 were covered. In order to measure the efficacy of the Brigada Digital de Salud, additional indicator should be used. Table 2 shows that 49,45 % of topics were related to Covid-19 vaccination: COVID Boosters 9 13 22 (6.01%) COVID Vaccination – Child/Adolescent 30 10 40 (10.92%) COVID Vaccination – General 81 11 92 (25.14%) COVID Vaccination – Safety/Efficacy 14 13 27 (7.38%). Therefore, the uptake of Covid-19 vaccination by the US spanish population could be a good indicator the Brigada efficacy.
RE: Thank you for your review of this manuscript. To clarify, the purpose of this article was not to assess efficacy of Brigada Digital. For this reason, we do not present measures of vaccine uptake or intentions. Given the scale and reach of this campaign intervention, using U.S. population-based measures for vaccination would overstate this intervention’s impact. For this reason, we have not included this as an outcome in this paper. We will be submitted a follow-up paper for publication that presents outcome measures for vaccine intentions and uptake. For the current paper, it is a process evaluation, so we include measures of reach and engagement of audience members.
Reviewer 2 Report
This paper is a detailed report of a Spanish language CHW project during COVID-19. The published literature does not have many papers on working with CHWs on health information products, especially on cultural adaptation, and this paper provides new information about how to work with CHWs to develop and distribute messages for specific audiences. The paper appears to have 3 different aims, however, which muddies its success as a research report and indicates there are 3 papers merged into 1. The first aim seems to align with the intro and methods sections and describes a community-based COVID-19 vaccine project with CHWs. The process details about the collaboration and message development are useful for other groups that want to do similar community projects with CHWs, but this reads like an excellent field report rather than a research study. Lines 67-68 says this is a report of a process and there isn't an explicit research question(s). The second aims seems to be reporting results of CHW distributed social media messages, which could have been stated as a research question but wasn't (e.g. how effective are Spanish-speaking CHWs in distributing COVID-19 social media messages through personal and organizational channels?). Moreover, the total number of posts (366) seems very low for a 2 year period, especially with 10 CHWs. It also doesn't add up with the post frequency schedule in the methods section. The results section includes a lot of counting of basic activities without any denominators or comparisons to other similar efforts or campaigns (how many people should we have expected the CHWs to reach in 2 years with the projected posting frequency scheduled?), while the potentially interesting results in section 3.3 get comparatively little attention. The third aim seems to be a comparison of different channels and formats for distributing Spanish language COVID-19 vaccine messages. The mixing of CHWs and social media both text and audiovisual, CHWs as guests on a radio program, and observations about different social media channels indicate that the authors think there is something interesting about channel and format differences as well. The Discussion section is long and tries to comment on these 3 distinct but related (by virtue of being part of the same project) aims of the paper, which is confusing. I finished the paper not completely sure what the authors think the significance of their findings is, and their recommendations seem mainly practical about how to work with CHWs and social media. In sum, this paper has valuable information about 3 different aspects of an important topic (CHWs as content collaborators and distributors in various formats and channels), but the information should be presented in 3 different papers that are likely more best practice oriented than original research study.
Author Response
Reviewer 2 Comments and Suggestions for Authors:
- This paper is a detailed report of a Spanish language CHW project during COVID-19. The published literature does not have many papers on working with CHWs on health information products, especially on cultural adaptation, and this paper provides new information about how to work with CHWs to develop and distribute messages for specific audiences. The paper appears to have 3 different aims, however, which muddies its success as a research report and indicates there are 3 papers merged into 1. The first aim seems to align with the intro and methods sections and describes a community-based COVID-19 vaccine project with CHWs. The process details about the collaboration and message development are useful for other groups that want to do similar community projects with CHWs, but this reads like an excellent field report rather than a research study. Lines 67-68 says this is a report of a process and there isn't an explicit research question(s). The second aim seems to be reporting results of CHW distributed social media messages, which could have been stated as a research question but wasn't (e.g. how effective are Spanish-speaking CHWs in distributing COVID-19 social media messages through personal and organizational channels?).
RE: Thank you for your careful review of this manuscript. We have edited the narrative to clarify that this is a process evaluation of the campaign intervention, a research inquiry, and we have added our research questions:
- What level of audience reach can be achieved using the Brigada Digital model for digital health promotion? and
- What level of audience engagement can be achieved using this model, and what post features are most engaging for a general audience of Spanish-speaking Latino adults?
- Moreover, the total number of posts (366) seems very low for a 2 year period, especially with 10 CHWs. It also doesn't add up with the post frequency schedule in the methods section.
RE: We have updated the post frequency schedule to say that the posts disseminated per week was an average of 2-3 posts, given the variability from week to week. Also, we would like to emphasize that the CHWs themselves were not directly responsible for creating the content for posts. This is something we do suggest in the Discussion as a next step for this line of research – to build community capacity and train CHWs to do this. As discussed in the Methods narrative, this content development process was centralized, and content was created by a small research team. While the CHWs provided ideas for posts, guided the selection of topics to address, and feedback on posts, the primary role of the CHWs was to disseminate the posts and discuss them with members of their social media networks. We added text to indicate that CHWs spent about 4-5 hours per week dedicated to their role on this project. We have also added additional detail to the narrative to clarify the capacity of the content development team, including that it consisted of 3 research team members (and it was actually quite a lot of work to develop 251 original posts (366 was the number of times a COVID-related topic was covered across the 251 posts since 115 of the posts covered more than 1 topic) with only three individuals working on this project on a part-time basis giving the evolving messaging, extensive research required to develop messaging and fact-check, and to develop the graphics and multimedia elements).
- The results section includes a lot of counting of basic activities without any denominators or comparisons to other similar efforts or campaigns (how many people should we have expected the CHWs to reach in 2 years with the projected posting frequency scheduled?), while the potentially interesting results in section 3.3 get comparatively little attention.
RE: Given the paucity of studies using a similar community-based digital health promotion approach with which we can compare this study’s results, this was exactly one of the questions we were trying to answer: how many people can be reached using an effort of this scope and scale and using this CHW digital outreach approach? Also, will the audience engage with this content (a preliminary indicator of the content’s appropriateness, appeal, and/or utility). The intention of this manuscript was to fill this gap in the literature that Reviewer 2 so keenly identifies, so that other studies may become familiar with and replicate this overall approach and to serve as a comparison for future studies testing this approach. We have revised the Discussion section to position this study’s findings with the literature as best we could given the lack of an existing literature base for this specific approach.
- The third aim seems to be a comparison of different channels and formats for distributing Spanish language COVID-19 vaccine messages. The mixing of CHWs and social media both text and audiovisual, CHWs as guests on a radio program, and observations about different social media channels indicate that the authors think there is something interesting about channel and format differences as well.
RE: It is possible that we did not adequately convey the intentions for including specific results. This has hopefully been clarified through the inclusion of research questions and the restructuring of both the results and discussion sections around those research questions. The purpose of including results for post topics and media/formats was to first characterize the Brigada Digital content that was disseminated, and to lay the groundwork for reporting results for the post topics/formats/features that were most engaging for this audience. Table 7 results are presented to show the post attributes that we reviewed in order to identify patterns of similarity (in terms of content, format, features) across the most engaging posts. Identifying these patterns aids in the formation of hypotheses as to why these posts were more engaging than others. For example, most of the top performing posts were videos, suggesting that videos were among the more engaging content and future campaign efforts should consider this finding regarding format. Similarly, posts that portrayed a well-known music artist or influencer resulted in more engagement, suggesting these as a potentially promising strategies should be further used and tested. Also, the health promotion and crisis communication literatures discuss the importance of messenger credibility and trustworthiness, although this could include many types of messengers and is dependent on the audience. From this standpoint, we created content that included CHWs as interviewed guests on a well-known Spanish radio show with a trusted physician host, and study results showed that this kind of content achieved more audience engagement (which is a preliminary indicator of appeal and/or utility, among other things). Describing the ways that we operationalized the communication strategy of ‘messaging by a credible source’ is useful for replication purposes and also for exploring reasons that may explain the audience engagement that was observed.
- The Discussion section is long and tries to comment on these 3 distinct but related (by virtue of being part of the same project) aims of the paper, which is confusing.
Re: We have organized the Discussion around our added research questions, and we have removed some text. We hope that this structure facilitates a more organized and compelling discussion overall.
- I finished the paper not completely sure what the authors think the significance of their findings is, and their recommendations seem mainly practical about how to work with CHWs and social media. In sum, this paper has valuable information about 3 different aspects of an important topic (CHWs as content collaborators and distributors in various formats and channels), but the information should be presented in 3 different papers that are likely more best practice oriented than original research study.
RE: Thank you for this comment. Positioning the results of this study within the current literature is trickier given the novelty of this digital health promotion approach and campaigns of this more modestly-sized scale, but we find that the lack of research in this area, identified by Review 2, is all the more reason to pursue publication of this work. One aim of this study is, indeed, to detail the overall approach for intervention researchers who would like to replicate it, as well as to provide the results of a campaign process evaluation, which is a perfectly acceptable form or research inquiry and also a contribution (especially since process evaluations are also often lacking in the literature). In order to clarify the significance of this study’s findings, we have positioned the results for reach and engagement within the closest literature that we can find to serve as a comparison. Specifically, we have discussed findings for reach and engagement with other studies examining these outcomes. We have also expanded upon the significance of study findings. We hope that Reviewer 2 agrees that these revisions help to convey the significance of research findings.
Reviewer 3 Report
I was invited to revise the paper entitled "Assessing Brigada Digital de Salud Audience Reach and Engagement: A Digital Community Health Worker Model to Address COVID-19 Misinformation in Spanish on Social Media". The present study aimed to describe the the process of developingthe Brigada Digital model over a 2-year period. It aimed to address the proliferation of COVID misinformation in Spanish on social media by leveraging community health workers as trusted sources of health information. Totally, 10 HCWs were enrolled and trained to diseminate via social media information about covid-19 vaccination. In addition, enrolled HCWs engaged with network members to answer questions and to expand their digital network.
In methods sections Authors described the messaging strategies of the model.
The topic is interesting and the paper described an interesting public health project to face off the covid-19 misinformation.
Minor observations:
- Table 5: Authors should report also absolute number with percentage;
- Among discussions, Authors should discuss the topic of vaccine hesitancy among healthcare workers;
- Authors should compare their project with similar study condicted in other countries.
Author Response
Reviewer 3 Comments and Suggestions for Authors:
- I was invited to revise the paper entitled "Assessing Brigada Digital de Salud Audience Reach and Engagement: A Digital Community Health Worker Model to Address COVID-19 Misinformation in Spanish on Social Media". The present study aimed to describe the process of developing the Brigada Digital model over a 2-year period. It aimed to address the proliferation of COVID misinformation in Spanish on social media by leveraging community health workers as trusted sources of health information. Totally, 10 HCWs were enrolled and trained to disseminate via social media information about covid-19 vaccination. In addition, enrolled CHWs engaged with network members to answer questions and to expand their digital network. In methods sections Authors described the messaging strategies of the model. The topic is interesting and the paper described an interesting public health project to face off the covid-19 misinformation. Minor observations:
- Table 5: Authors should report also absolute number with percentage;
RE: Per Reviewer 3’s suggestion, we have added absolute numbers to Table 5.
- Among discussions, Authors should discuss the topic of vaccine hesitancy among healthcare workers;
RE: Discussing vaccine hesitancy among healthcare workers is beyond the scope of this paper given that the intervention’s intended audience is not healthcare workers. We did not collect data or discuss any results related to vaccine hesitancy among healthcare workers. There are other articles published in the Special Issue that have a primary purpose of examining the issue of hesitancy of healthcare workers, and are able address this topic more adequately. Given these reasons, we acknowledge Review 3’s suggestions, but would respectfully prefer to not add a topic to the discussion that we could not adequately discuss as it pertains to the goals of this study.
- Authors should compare their project with similar study conducted in other countries.
RE: We have revised the Discussion section to better position the findings of this study within the literature. As noted by another reviewer, the published literature does not have many papers on this topic, and we are somewhat limited by a lack of studies in the U.S. that have used a digital health promotion approach with Community Health Workers. Studies from other countries may not always serve as good comparisons for a few reasons, principally among them is that Latino, Spanish-speaking individuals in the U.S. constitute a language minority population that experienced inequitable access to quality COVID-19 information during the pandemic. We have added some discussion with a comparison to other studies. We hope that Reviewer 3 agrees that the inclusion of studies that examine similar process evaluation outcomes for social media campaigns have better conveyed how our study’s results for reach and engagement compare.
Round 2
Reviewer 2 Report
The authors were responsive to reviewers' comments, and their edits resolve most of my questions about the manuscript's aims. The additions of other studies are helpful in contextualizing the methods and results. What I missed on first read and saw on the second read is that number of social media accounts and the audience size for Huerta's radio program are missing. The authors could further provide readers a sense of project scope by adding the number of accounts used for each platform and the size of the listener audience for the radio program(s). There are several places in the manuscript where the authors discuss the use of Brigada Digital accounts and CHWs' accounts but I don't have a sense of how many accounts were used to generate the numbers of reported posts and engagements. Also, did La Clinica or Proyecto Salud use their organizational accounts to distribute posts or advertise the radio programs?
Line 82: sentence is missing a verb
Line 450: grammar issue
Line 492: "not to mention" makes sentence intro awkward
Author Response
Dear Reviewers:
Thank you again for your helpful comments on this manuscript, which we have carefully considered. Below, we detail how the text has been revised accordingly or provide additional clarifications. Revisions in the manuscript are indicated with red text.
Reviewer 2:
The authors were responsive to reviewers' comments, and their edits resolve most of my questions about the manuscript's aims. The additions of other studies are helpful in contextualizing the methods and results. What I missed on first read and saw on the second read is that number of social media accounts and the audience size for Huerta's radio program are missing. The authors could further provide readers a sense of project scope by adding the number of accounts used for each platform and the size of the listener audience for the radio program(s).
RE: Thank you for bringing this to our attention. We have inserted information for the number of accounts used in addition to the 3 main Brigada Digital accounts (lines 285-286). While we do not have more precise information about the exact audience size for Dr. Huerta’s program, we do provide information about the market where the program airs, (the Washington, DC metropolitan area), and the estimated population size for Latinos (lines 116-117). This program has been on the air for decades, so we estimate that it has considerable, loyal following.
There are several places in the manuscript where the authors discuss the use of Brigada Digital accounts and CHWs' accounts but I don't have a sense of how many accounts were used to generate the numbers of reported posts and engagements.
RE: We have clarified in the text that while there were multiple social media accounts of individual Brigada Digital CHWs, there were only 3 main Brigada Digital accounts, one for each platform. All original posts were published first through the main 3 accounts, and then shared by CHWs to their networks. All metrics reported (reach, engagement, etc.) are from the 3 main accounts. Edits can be found on lines 332, 352, 358, and 369.
Also, did La Clinica or Proyecto Salud use their organizational accounts to distribute posts or advertise the radio programs?
RE: We have included the detail that Clinica del Pueblo did share our posts with their audience on Facebook.
Line 82: sentence is missing a verb
RE: This has been corrected.
Line 450: grammar issue
RE: This has been corrected.
Line 492: "not to mention" makes sentence intro awkward
RE: This has been corrected.
